# CRISPR mediated transactivation in the human disease vector *Aedes aegypti*

**Michelle Bui[1], Elena Dalla Benetta[1], Yuemei Dong[2], Yunchong Zhao[1], Ting Yang[1], Ming Li[1], Igor A. Antoshechkin[3], Anna Buchman[1¤], Vanessa Bottino-Rojas[4], Anthony A. James[4,5], Michael W. Perry[1], George Dimopoulos[2], Omar S. Akbari[1]***

**1** Department of Cell and Developmental Biology, School of Biological Sciences, University of California San Diego, San Diego, California, United States of America, **2** W. Harry Feinstone Department of Molecular Microbiology and Immunology, Bloomberg School of Public Health, Johns Hopkins University, Baltimore, Maryland, United States of America, **3** Division of Biology and Biological Engineering, California Institute of Technology, Pasadena, California, United States of America, **4** Department of Microbiology & Molecular Genetics, University of California Irvine, Irvine, California, United States of America, **5** Department of Molecular Biology & Biochemistry, University of California Irvine, Irvine, California, United States of America

¤ Current address: Verily Life Sciences, San Francisco, California, United States of America
* oakbari@ucsd.edu

**Data Availability Statement:** All plasmids and annotated DNA sequence maps are available at www.addgene.com under accession numbers: #183993, #100581, #184006, #184007, #120363, #190997. Raw sequencing data are available at

## Abstract

As a major insect vector of multiple arboviruses, *Aedes aegypti* poses a significant global health and economic burden. A number of genetic engineering tools have been exploited to understand its biology with the goal of reducing its impact. For example, current tools have focused on knocking-down RNA transcripts, inducing loss-of-function mutations, or expressing exogenous DNA. However, methods for transactivating endogenous genes have not been developed. To fill this void, here we developed a CRISPR activation (CRISPRa) system in *Ae. aegypti* to transactivate target gene expression. Gene expression is activated through pairing a catalytically-inactive ('dead') Cas9 (dCas9) with a highly-active tripartite activator, VP64-p65-Rta (VPR) and synthetic guide RNA (sgRNA) complementary to a user defined target-gene promoter region. As a proof of concept, we demonstrate that engineered *Ae. aegypti* mosquitoes harboring a binary CRISPRa system can be used to effectively overexpress two developmental genes, *even-skipped (eve)* and *hedgehog (hh)*, resulting in observable morphological phenotypes. We also used this system to overexpress the positive transcriptional regulator of the Toll immune pathway known as *AaRel1*, which resulted in a significant suppression of dengue virus serotype 2 (DENV2) titers in the mosquito. This system provides a versatile tool for research pathways not previously possible in *Ae. aegypti*, such as programmed overexpression of endogenous genes, and may aid in gene characterization studies and the development of innovative vector control tools.

## Author summary

The mosquito, *Aedes aegypti* is a major vector of arthropod-borne viruses such as Yellow Fever, Chikigunya, and Zika virus. To prevent the spread of disease, the ability to control and reduce *Ae. aegypti* populations are critical. Historically, this has been accomplished

NCBI Sequence Read Archive, at https://www.ncbi.nlm.nih.gov/bioproject/PRJNA851480.

**Funding:** This project has been funded in part by a DARPA Safe Genes Program Grant under contract number, HR0011-17-2-0047 and NIH awards, R01AI151004, DP2AI152071, and R21AI149161 awarded to OSA, and R01AI141532 awarded to GD. Our funders had no role in study design, data collection and analysis, decision to publish, or the preparation of this manuscript.

**Competing interests:** I have read the journal's policy and the authors of this manuscript have the following competing interests: O.S.A. is a founder of both Agragene, Inc. and Synvect, Inc. with equity interest. The terms of this arrangement have been reviewed and approved by the University of California, San Diego in accordance with its conflict of interest policies. The authors report no other conflict of interest.

using chemical insecticides. However, insecticides are expensive, non-specific, and have led to the development of resistance within some mosquito populations. Therefore, alternative methods have been sought after. In particular, the development of genetic-based control strategies have been of focus. Current genetic-based tools for *Ae. aegypti* depend upon disrupting genes that can lead to a reduction of mosquito population size or the prevention of disease transmission. Other tools focus on inserting foreign DNA that can encode mechanisms for disease resistance or encourage continued inheritance of foreign DNA within populations. Despite all these methods there has yet to be a method of inducing expression of the mosquitoes own genes within their own genome. Here we generated a CRISPRactivation (CRISPRa) system in *Ae. aegypti* and demonstrated its efficacy through activating the expression of the developmental genes *even-skipped (eve)* and *hedgehog (hh)* as well as regulator of an immune pathway *AaRelI*.

## Introduction

The yellow fever mosquito, *Aedes aegypti*, is a competent vector of arboviruses including chikungunya, dengue, and Zika [1–3]. Their vectorial capacity, desiccation-tolerant eggs, adaptability to a range of climates and anthropophilic behavior have enabled them to become and remain an increasing burden to human welfare [4–6]. As global temperatures increase, the spatial distribution and range of *Ae. aegypti*, as well as the pathogens they transmit, continue to expand [7]. Historically, insecticides have been the major tool for reducing mosquito populations to control the spread of mosquito-borne diseases. However, as mosquito populations continue to thrive, and evolve resistance to insecticides, alternative control measures are of utmost demand. In particular, strategies centered around genetic manipulation have become a major focus for novel genetic-based tool development.

In tandem with improved and expanding assemblies of the *Ae. aegypti* genome and various transcriptomes [8–10], pivotal tools in the development of genetic-based vector control strategies have been the reduction of transcript levels using RNAi [11], germline transformation using transposable elements [12], and programmable DNA targeting-using Clustered Regularly Interspaced Short Palindromic Repeat (CRISPR) [13–15]. For example, RNAi and other small RNAs have been instrumental in generating knock-down phenotypes within the mosquito [16–18] as well as a method for reducing viral transcript numbers [18–22]. Moreover, CRISPR/Cas9 has been pivotal for site-directed mutagenesis and site-specific recombination to become more efficient, precise, and accessible [14,23]. For example, CRISPR/Cas9 mutagenesis has been utilized in *Ae. aegypti* for furthering the understanding of various factors of mosquito biology, such as sex determination [24,25], olfaction [26], behavior [27] and development [18–22,28,29]. In addition to site-directed mutagenesis, CRISPR/Cas9 has been used to integrate desired DNA sequences into the mosquito genome through Homology Directed Repair (HDR)[23]. In conjunction, these tools have been used to develop numerous vector control strategies in *Ae. aegypti* including those based on conferring dengue and Zika virus resistance [18,30], homing based gene drives [31,32], and recently the precision-guided sterile insect technique (pgSIT) [15]. Although these tools have been instrumental to the expansion of vector control methods, additional tools are needed to explore a broader range of options for vector control. Tools like the binary UAS/GAL4, and the tetracycline-repressible transcriptional activator (tTA) systems, can be used to express transgenes of interest mosquitoes [33,34] However, the main drawbacks of both approaches is that they cannot activate gene expression at their endogenous locus. Thus far, the ability to specifically induce endogenous

gene overexpression in insect vectors of disease does not exist, and therefore providing these tools will enable precise gene modulation in basic studies of functional gene activity.

Recently, researchers have engineered a CRISPR-based tool able to function as a programmable transcription factor for transactivating the expression of target genes [35,36]. The system, known as CRISPR activation (CRISPRa), utilizes a nuclease-deactivated, or dead, Cas9 (dCas9) able to bind to a target locus with the aid of a complementary spacer sequence called a small guide RNA (sgRNA) [35,37]. However, unlike Cas9, dCas9 contains two mutations that disable its endonucleolytic activity, thus preventing cleavage of DNA [37]. Interestingly, when fused with the tripartite transcriptional activators VP64-p65-Rta (VPR), dCas9 can efficiently recruit transcriptional machinery to a promoter region by mimicking the natural cooperative recruitment process of transcription initiation [35]. Consequently, dCas9-VPR is able to transactivate the expression of a specific endogenous gene when guided to the promoter region of the gene of choice [35]. Furthermore, dCas9-VPR can be directed to nearly any DNA sequence by an sgRNA, requiring only a short protospacer adjacent motif (PAM) site 5'-NGG-3' proximal to the target. With the ability to bind and recruit transcription factors, CRISPRa has previously been utilized to effectively upregulate target genes in human cells, *Bombyx mori* cell lines, and in *Drosophila melanogaster* [35]. Endogenous transactivation is a desirable goal for a number of basic and applied research purposes. In *D. melanogaster*, CRISPRa has been leveraged to engineer synthetic species by creating stable reproductive barriers that produce an artificial selection pressure and drive genes through a population in a reversible manner [38]. Engineering this technique in other species, including mosquitoes, could provide a safe platform for modification of wild populations. Additionally, functional studies involving upregulation of key regulatory genes can be performed to study a wide range of mosquito pathways including the role of immune response after virus infection. However, *in vivo* use of this tool is limited and it has not been demonstrated yet in any disease vectors.

Here we have generated the first CRISPRa system in *Ae. aegypti*. To determine the efficacy of our system we targeted the expression of two conserved developmental genes, *even-skipped* (*eve*, AAEL007369) and *hedgehog* (*hh*, AAEL006708) that play instrumental roles in the spatial and temporal control of embryonic developmental patterning. *Eve* is involved in the development of odd- and even-numbered parasegments [39], whereas *hh* signaling plays numerous roles such as the development of segment polarity and various organs [40]. Following transactivation of these genes using our CRISPRa system, we quantified targeted overexpression using both qPCR and RNA sequencing. In addition, we observed phenotypic changes such as lethality and spatial cellular ectopic expression visualized by *in situ* hybridization. Similarly, as a proof of principle we applied the CRISPRa system to transactivate the positive transcriptional regulator gene of Toll immune pathway, known as *AaRel1 or Rel1* (AAEL007696). *Rel1* plays a central role as a positive transcriptional factor in antiviral and anti-fungal defenses. The activation of the Toll pathway can be monitored through the transcriptional activation of the *Rel1* transcription factor or down-regulation of the negative regulator *Cactus*. Transgenic overexpression of *AaRel1*, or RNA interference (RNAi)-mediated silencing of the negative regulator *Cactus*, have demonstrated potent antiviral and anti-fungal roles of the Toll immune pathway [41–44]. Here we have shown the CRISPRa-mediated transactivation of *AaRel1* resulted in a similar level of suppression of viral (DENV2) infection in *Ae. aegypti* mosquitoes similar to what was observed in the *Cactus* gene silenced mosquitoes [43]. Transactivating specific immune genes not only could inform us on their functional role in response to viral infection, but the CRISPRa technology also can be a powerful tool for investigating gene function in *Ae. aegypti*.

## Results

### Generation of CRISPRa transgenic lines in *Ae. aegypti*

To engineer a CRISPRa system in *Ae. aegypti*, we generated a transgenic mosquito line endogenously expressing a catalytically dead Cas9 (dCas9) fused with a transcriptional activator, VP64-p65-RTA (VPR), as described by previous CRISPRa systems created in *D. melanogaster* [45,46]. Expression of dCas9-VPR was driven by a *polyubiquitin* (PUb) promoter known to be active at relatively high expression levels within a variety of cell types and developmental stages [47]. PUb was chosen to generate the constitutive expression of dCas9-VPR and thus promote transactivation of the target genes throughout the mosquito body and developmental stages. (**Fig 1B**). In addition, we generated three transgenic lines expressing sgRNAs targeting either *even-skipped (eve)*, *hedgehog (hh)*, or *AaRel1*. To enable robust target gene transactivation, the sgRNA target sites were designed from sequences within 0–250 base-pairs (bp) adjacent to the 5'-end (upstream) the transcription start site (TSS) (**Fig 1A**). Moreover, to increase gene activation efficacy, each sgRNA line encodes two or four different sgRNAs targeting different locations within the 250bp upstream of the TSS of the gene of interest (**Fig 1B**). With the use of multiplexed sgRNAs, gene activation is promoted synergistically by multiple dCas9VPR/sgRNA binding events at the proximal region of the promoter [48]. Expression of sgRNAs was ubiquitously driven by either U6a or U6b promoters [31]. Fluorescent markers were encoded to confirm integration of the transgenes into the mosquito genome as well as tracking inheritance of the constructs (**Fig 1B and 1C**).

### CRISPRa induces target gene overexpression

To confirm and quantify CRISPRa-mediated transactivation, we performed genetic crosses between PUb: dCas9-VPR/+ males and U6:sgRNA$^{eve}$/+ or U6:sgRNA$^{hh}$/+ females. Resulting progeny were collected to quantify expression and transcript abundance of the target genes and observe phenotypes resulting from overexpression (**Fig 1A**). Total RNA was extracted first from 24h post oviposition eggs from insects transheterozygous for dCas9-VPR and U6-sgRNA and from controls, including the two parental lines and the wild-type line (Liverpool strain). Transcript levels for *eve* and *hh* were measured using both quantitative real-time PCR (qPCR) and RNA transcriptome sequencing (RNAseq). For both genes, qPCR analysis revealed significant increases in transcript abundance resulting from transactivation. Specifically, *eve* and *hh* showed a 22.8- and 8.20-fold increase, respectively, compared to wild-type and control samples (**Fig 2B**).

RNAseq and qPCR were performed to determine the effect of *eve* and *hh* overexpression on other related developmental genes and screen for possible off-targeting effects (**Fig 2C** and **S1**–**S6** **Tables**). Genome-wide transcriptome analysis confirmed the qPCR overexpression data. Both *eve* and *hh* showed increased TPM (Transcripts per million) values only in transheterozygous individuals. *eve* had an abundance value of 91.52 TPM, corresponding to a 5.7-fold increase, in transheterozygotes, whereas *hh* has an abundance value of 11.03 TPM, representing a 2.8-fold increase in transheterozygotes compared to controls (**Fig 2C** and **S1**–**S6** **Tables**). In addition, *hh* transheterozygous eggs also have a higher *eve* abundance value of 38.42 TPM, a 3.5-fold increase in transheterozygous individuals compared to the control lines (**Fig 2C**). The levels of upregulation also can be observed with the integrative genomics viewer (IGV) where a larger number of reads aligned to both *eve* and *hh* in the transheterozygous samples compared to the controls (**S1 and S2 Figs**). Taken together, both qPCR and RNAseq data indicate robust and programmable target gene transactivation in transheterozygotes.

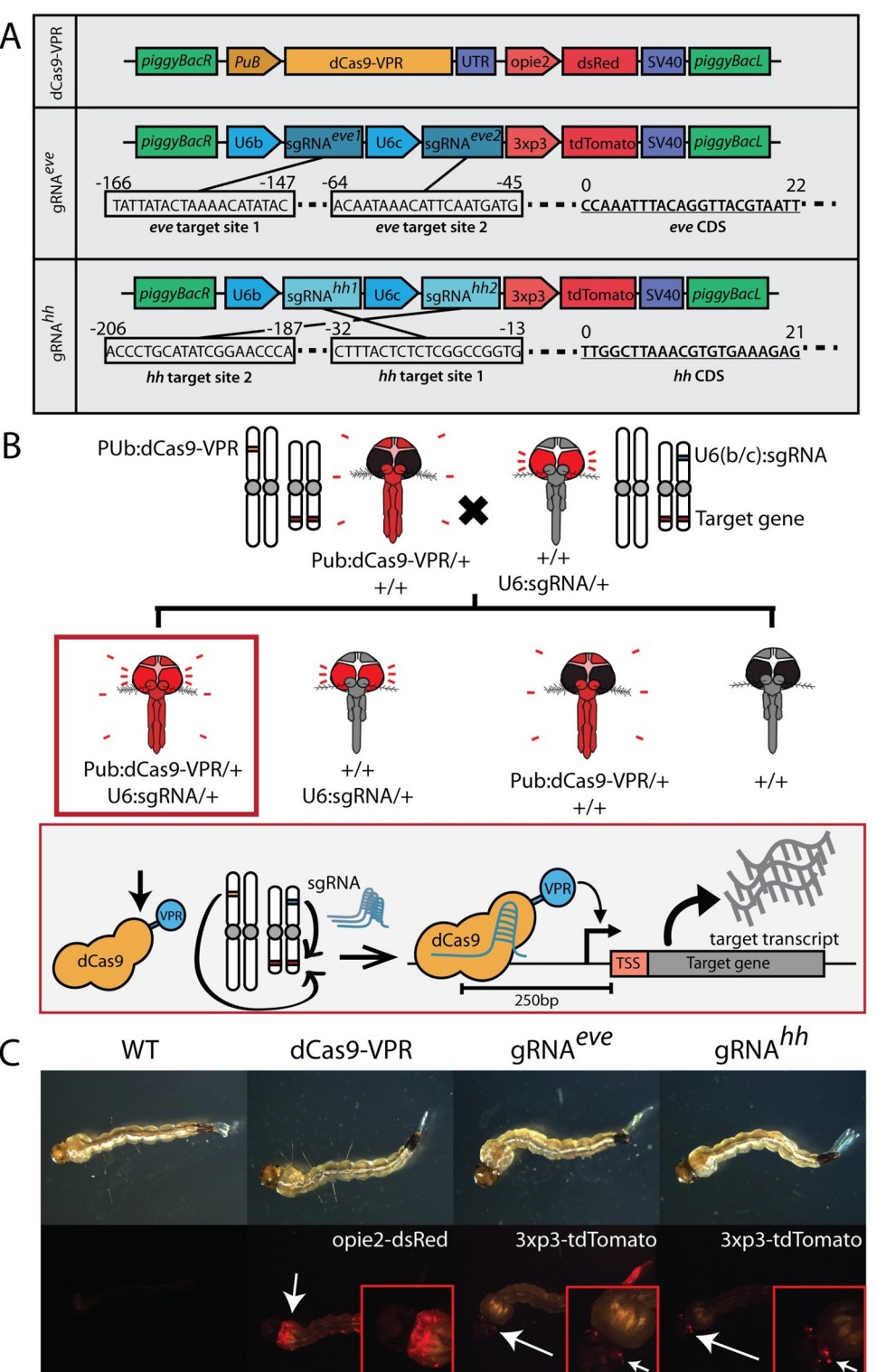

**Fig 1. CRISPRa system transgenic line design, genetic crosses, and marker expression in *Ae. aegypti*. A)** A binary CRISPRa system was designed using two separate transgenic *Ae. aegypti* lines that when crossed, result in transheterozygous individuals expressing both dCas9-VPR and sgRNAs and have upregulated expression of the target gene. **B)** The transgenic lines used in this study include one line expressing dCas9-VPR under a polyubiquitin promoter, a sgRNA line targeting *eve*, and a sgRNA line targeting *hh*. The sgRNA lines were designed to express two distinct sgRNAs targeting the same promoter region of the respective target gene. **C)** dCas9 and sgRNA lines were marked with opie2-dsRed and 3xP3-tdTomato, respectively.

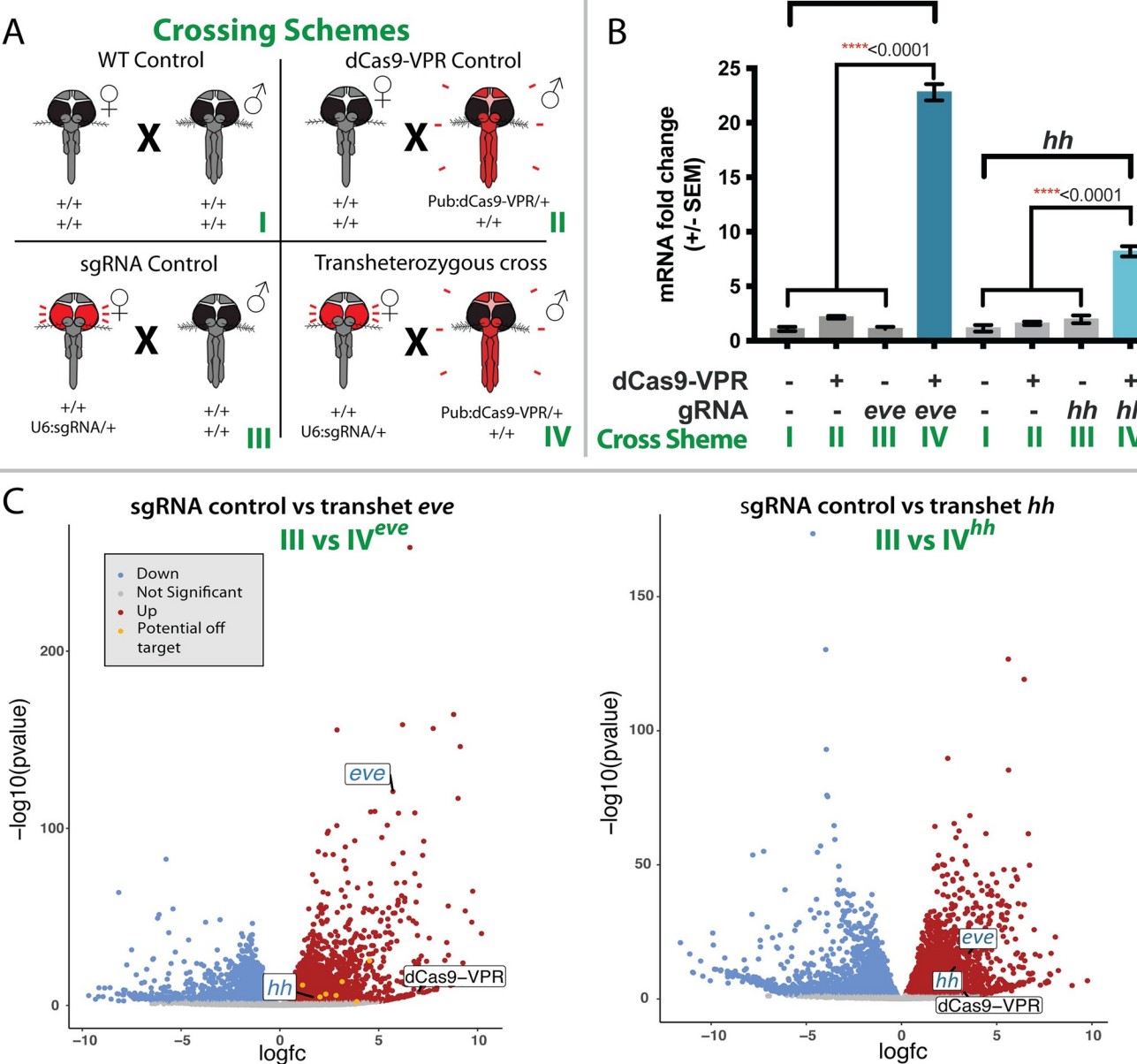

**Fig 2. Quantified overexpression of CRISPRa transactivation. A)** Multiple crosses were performed in order to determine if the CRISPRa system could transactivate target genes. Crosses included a wild-type control (+/+ ♀ X +/+ ♂, I), dCas9-VPR Control (+/+ ♀ X Pub:dCas9-VPR/+ ♂, II), sgRNA control (+/U6:sgRNA ♀ X +/+ ♂, III), and a transheterozygous cross (+/U6:sgRNA ♀ X Pub:dCas9-VPR/+ ♂, IV). **B)** Fold change of expression of targeted genes from CRISPRa transactivation was measured and calculated using qPCR. Differences in mRNA fold change between control crosses and transheterozygous crosses were calculated using a one-way ANOVA and Tukey's multiple-comparison test, ****$p < 0.0001$ ($F_{3,8} = 659.7$; $P = 6.46e-10$; $F_{3,8} = 82.67$; $P = 2.32e-06$ for *eve* and *hh* respectively). **C)** Volcano plots of the RNA sequencing data were to compare expression levels of *eve* and *hh* as well as other genes potentially affected. Down-regulated genes are colored blue while upregulated genes are in red. Potential off-target genes are in yellow.

## Overexpression of *eve* and *hh* generated additional transcriptomic changes

We performed a differential expression analysis using the RNAseq data to search for genome-wide effects that could result from target gene overexpression. Differential expression comparisons, between transheterozygous individuals with *eve* transactivated and control lines, identified ~43% of the total genes with significant transcript accumulation level changes

(FDR<0.05) (**Fig 2C** and **S3**, **S5** and **S7** **Tables**). Approximately 2000 of these genes had significant increases in transcript abundance of more than 2-fold (Log Fold Changes > 2 and FDR < 0.05). Among these upregulated transcripts, genes encoding fibrinogen, centrin, lipase, and an unspecified protein orthologous to the *Ae. albopictus* PIWI gee product showed the highest upregulation (**Fig 2C**). Similarly, the differential transcript abundance profile in *hh*-transactivated individuals also identified ~41% of the total genes with significant transcript level changes (FDR < 0.05). As in the *eve*-transactivated individuals, ~2200 of those genes displayed more than two-fold changes in abundance. Among the most differentially expressed genes that showed an increase in abundance, the most upregulated genes were chaperonin and steroid dehydrogenase. In contrast, the most significantly downregulated genes were ribosomal proteins from both the 40S and 60S subunits (**Fig 2C**). Taken together these results support the conclusion that the effect of *eve* and *hh* overexpression results in genome-wide differential gene expression.

To determine if dCas9-VPR has off-target activation effects as described previously [49,50], potential off-target binding sites were predicted bioinformatically using an optimal CRISPR target finding software [51,52]. We then analyzed each of the predicted off-target binding sites as described in [45] and to determine if any closely-linked genes were upregulated in our RNAseq experiments A total of 85 potential off-target sites were detected among all the sgRNA target sites (**S8** and **S9** **Tables**). The majority of off-target sites reside far from functional loci, in the intronic region, or in the proximity of genes that do not show any differential regulation, based on the RNA sequencing information (**S8** and **S9** **Tables**). However, eight off-target sites related to *eve* sgRNAs are localized upstream of genes that were differently expressed in our analysis, with significant fold changes higher than two-fold (**Fig 2C** and **S8** and **S9** **Tables**). Not surprisingly, there are some important developmental transcription factors among those genes, including *scabrous* [53] and *nk* homeobox genes [54], indicating that their upregulation is likely mediated by *eve* overexpression rather than off-targeting (**S8** and **S9** **Tables**). No potential off-target sites related to *hh* sgRNAs showed significant transactivation. These results indicate that the probability of off-target transactivation with dCas9-VPR is low. Nonetheless, the genome-wide differential gene expression is likely either a direct or indirect effect of *eve* and *hh* transactivation, confirming the ability of using this tool to induce an efficient and measurable overexpression.

## Transactivation results in lethality

Transheterozygous progeny were screened for hatching rate and visible morphological phenotypes potentially induced by CRISPRa-mediated transactivation. *eve* and *hh* transactivation resulted in varied rates of embryonic lethality (**Fig 3B**). Embryonic lethality rates were calculated by comparing the proportion of transheterozygous progeny resulting from heterozygous CRISPRa parents with the expected proportion, 25%, that would be seen with Mendelian inheritance if there was no impact. Interestingly, rates of embryonic lethality also were affected by the paternal or maternal lineage of CRISPRa elements. For example, when transactivating *eve*, if dCas9-VPR is inherited by the heterozygous males and the sgRNA by heterozygous females, only 12% of the offspring resulted in transheterozygotes. This represents a 52% reduction from the expected mendelian inheritance. In contrast, the reciprocal crosses in which the dCas9-VPR element is inherited maternally and the gRNA paternally, ~21% of the transheterozygotes survived, indicating 16% reduction from the expected value. Similarly, when transactivating *hh*, when the dCas9-VPR is inherited from the male, only 1% of the transheterozygous individuals hatched to larvae, indicating a more severe phenotype when *hh* is transactivated. If the dCas9-VPR is inherited through the heterozygous mother, an average of 11% (56%

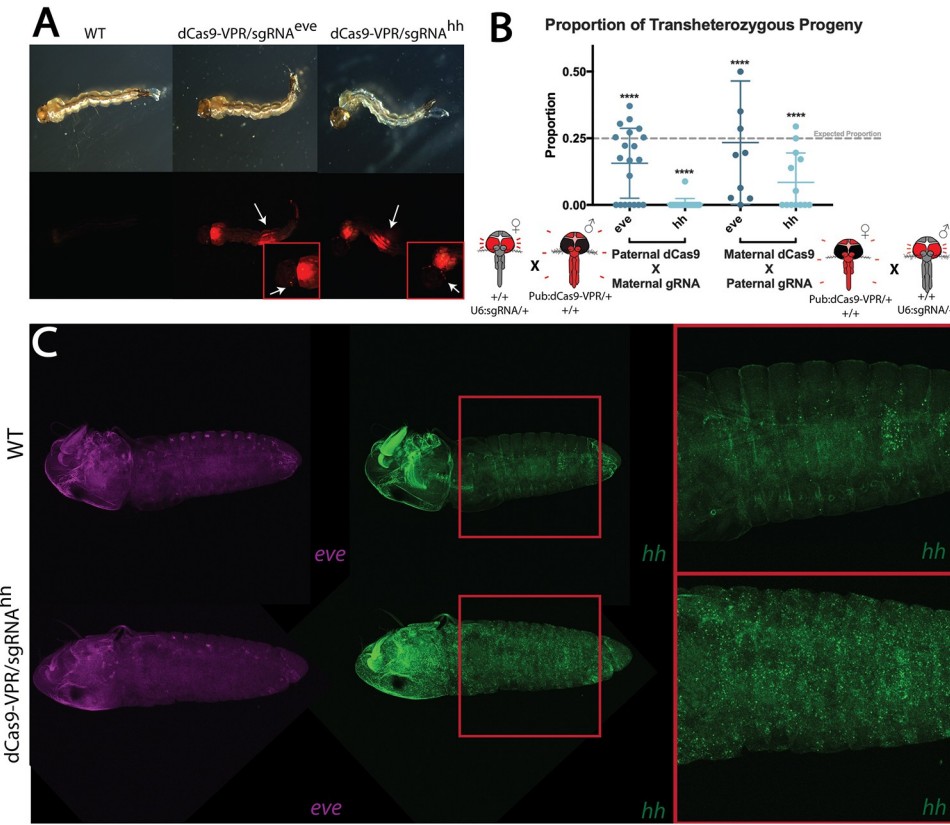

**Fig 3. Phenotypic observations of CRISPRa mediated transactivation of *eve* and *hh*. A)** Surviving transheterozygous progeny were collected and showed no significant external morphological differences to wild-type larvae. **B)** The proportions of transheterozygous progeny were calculated from single pair crosses between heterozygous dCas9-VPR and sgRNA parents. Rates of inheritance were compared to an expected rate of Mendelian inheritance, 25% transheterozygous individuals among total offspring, using a Chi-square test (****$p < 0.0001$). **C)** Further imaging of HCR *in situ* hybridization of *hh* probes showed upregulated and misregulated expression of *hh* compared to WT, while *eve* expression stayed relatively the same as that of WT.

reduction from expected) of the transheterozygotes survived to larval stage. Thus, transactivating either *eve* or *hh* resulted in higher rates of lethality when dCas9-VPR was inherited paternally and sgRNAs were inherited maternally. Furthermore, upregulating *hh* expression resulted in a higher frequency and higher severity of lethality then targeting *eve*. It is not surprising that their upregulation resulted in high lethality given the role these two genes play in development.

We collected surviving transheterozygous larvae to screen microscopically for visible morphological phenotypes that could be linked to ectopic expression of the target genes. Both PUb:dCas9-VPR/U6-sgRNA*eve* and PUb:dCas9-VPR/U6-sgRNA*hh* transheterozygotes did not have any obvious differences in morphology compared to WT larvae (**Fig 3A**). In addition, only larvae with *eve* transactivated survive to adulthood, and interestingly those larvae did not show a significant level of upregulation when eve transcript level was tested with qPCR (S3 Fig). Hybridization *in situ* was conducted on PUb:dCas9-VPR/U6-sgRNA*hh* embryos with probes designed for *eve* and *hh* transcripts to evaluate overexpression in embryos. Imaging showed increased accumulation of *hh* transcripts throughout the embryo and delayed development when compared to WT embryos at the same developmental time point (**Fig 3C**). A significantly increased abundance of *eve* was not seen in PUb:dCas9-VPR/U6-sgRNA*hh* embryos,

however a subtle difference in localization was observed. This could be an artifact from *hh* transactivation affecting downstream genes. The ectopic expression of *hh* confirms that the PuB promoter can be used to transactivate in cells throughout the embryo. Furthermore, the delayed development of the PUb:dCas9-VPR/U6-sgRNA$^{hh}$ embryos likely correlates with defects that would lead to the diminished numbers of transheterozygous progeny seen in our phenotypic screening crosses. Taken together, these results support the conclusion that our PUb:dCas9-VPR line is able to increase expression of target genes throughout the embryo.

## Overexpression of *AaRel1* results in virus suppression

The mosquito Toll and JAK-STAT immune signaling pathways and the siRNA immune pathway, play important roles in defending against arboviral infections [43,55,56] (reviewed in [57]). As a proof of principle, we used CRISPRa to transactivate *AaRel1*, an NF-κB Relish-like transcription factor that mediates the Toll immune pathway's antipathogenic action including the suppression of Dengue and Zika viruses [41–44]. First we generated a transgenic line expressing sgRNAs targeting the promoter region of *AaRel1* gene under the U6 promoter as described above (**Fig 4A**) (U6:sgRNA$^{rel1}$). To increase efficacy, the sgRNA line encodes four sgRNAs targeting four different regions within the promoter region of *AaRel1* (**Fig 4A**). It is known that CRISPRa applications benefit from multiplexing and that gene activation is promoted synergistically by multiple dCas9VR/sgRNA binding events at the proximal region of the promoter. For example, dCas12a-based activators with three gRNAs targeting a single gene in human cells increased gene expression by 9- to 40-fold relative to a single gRNA target [48]. The sgRNA target sites were designed from sequences within 0–250 base-pairs (bp) at the 5'-end of the transcription start site (TSS) to enable robust target gene transactivation. Expression of sgRNAs was driven ubiquitously by either U6a or U6b promoters [31]. Fluorescent markers were encoded to confirm integration of the transgenes into the mosquito genome as well as tracking inheritance of the constructs. Two lines harboring the same sgRNA sequences but with different insertion sites were created and used for subsequent analysis U6:sgRNA$^{rel1-A}$ and U6:sgRNA$^{rel1-B}$.

Crosses between PUb:dCas9-VPR/+ males and U6:sgRNA$^{rel1-A}$/+ females and PUb:dCas9-VPR/+ males and U6:sgRNA$^{rel1-B}$/+ females were performed to quantify CRISPRa-mediated transactivation and its effect on virus replication. Resulting progeny were collected to quantify expression and transcript abundance of the target genes and perform Dengue virus challenge (**Fig 4B**). *AaRel1* transcript levels were measured with qPCR from 11-day post-emergence adult transheterozygous females for both crosses and controls as described above. qPCR analysis of *rel1* resulted in a significant 18.02- and 11.78-fold increase in transcript levels, respectively, for both transactivated lines compared to their respective controls (**Fig 4C**). Non-significant difference in transcript abundance was observed between transheterozygous lines dCas9-VPR/U6-sgRNA$^{rel1-A}$ and dCas9-VPR/U6-sgRNA$^{rel1-B}$ (One-way ANOVA, Tukey's multiple-comparison test, $P = 0.224$).

To investigate the impact of transactivation of *AaRel1* on dengue virus serotype 2 (DENV2) infection, the transheterozygous lines dCas9-VPR/U6-sgRNA$^{rel1-A}$ and dCas9-VPR/U6-sgRNA$^{rel1-B}$, along with the siblings of the corresponding U6:sgRNA$^{rel1-A}$ and U6:sgRNA$^{rel1-B}$ mosquitoes as controls were orally infected with an artificial blood meal containing $10^7$ PFU/mL virus particles (**Fig 4D**).The viral infection intensity and prevalence in the midgut at 7 days post-infectious blood meal (PIBM) was determined through plaque assay. The median virus titers (infection intensities) in the mosquito midguts were reduced significantly by 3-fold and 2.5-fold, respectively, in the midgut tissues of both transactivated lines when compared to the corresponding controls (**Fig 4E**, Mann-Whitney test, $^{**} P < 0.01$). The infection

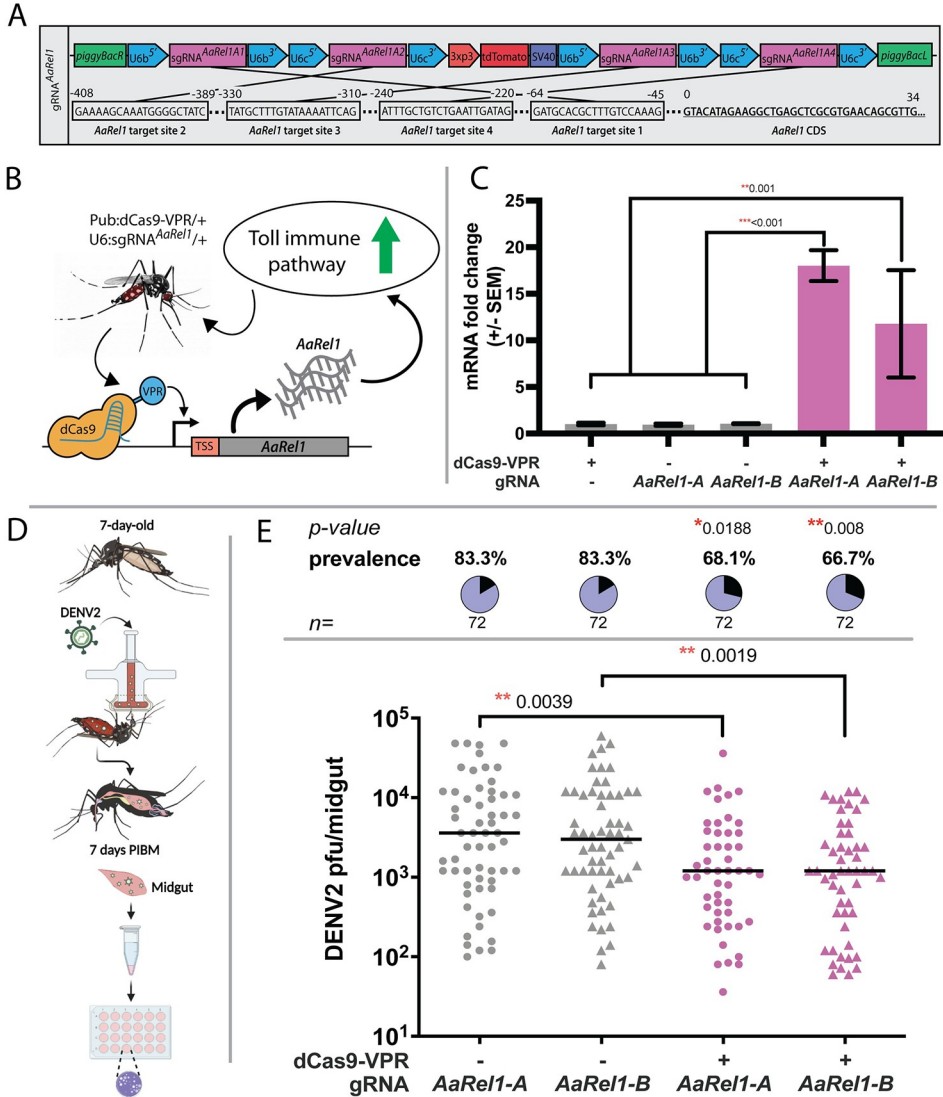

**Fig 4. CRISPRa mediated transactivation of Toll immune pathway, RT-PCR validation of *AaRel1* overexpression and suppression of viral infection. A)** Schematic representation of the sgRNA *AaRel1* construct used to create the U6: sgRNA-*AaRel1* lines. **B)** Schematic representation of the transgenic line expressing Pub:dCas9-VPR and the U6: sgRNA targeting the promoter region of *AaRel1* gene and inducing the activation of the Toll immune pathway. **C)** Quantification by qPCR of *AaRel1* gene expression in controls and transactivated lines. A one-way ANOVA and Tukey's multiple-comparison test was performed between transheterozygous crosses and controls. *$P < 0.05$, **$P < 0.01$, ***$P < 0.001$ **D)** Antiviral effect of CRISPRa mediated transactivation of *AaRel1* in the transheterozygous progeny. sgRNA-*AaRel1* line A and B (sgRNA-*AaRel1*-A and sgRNA-*AaRel1*-B, or named as U6:sgRNA$^{rel1-A}$ and U6: sgRNA$^{rel1-B}$) and transheterozygotes (dCas9-VPR/sgRNA-*AaRel1*-A and dCas9-VPR/sgRNA-*AaRel1*-B or named as dCas9-VPR/U6-sgRNA$^{rel1-A}$ and dCas9-VPR/U6-sgRNA$^{rel1-B}$) were orally infected with DENV2 as illustrated. **E)** The viral infection titers and infection prevalence of DENV2 were measured after midgut infection at 7 days post-infectious blood meal (PIBM). Plaque assays were used to determine viral titers and infection prevalence in individual mosquitoes with each dot representing the viral load from one midgut, and pie-chart indicating the infection prevalence. at At least three replicates were pooled for the statistical analyses. Horizontal black lines indicate the median of the viral loads. Considering the non-normal distribution of viral titers, median is used to describe central tendency and non-parametric Mann-Whitney test was used to compare median viral titers and Fisher's exact test to compare infection prevalence. * $P < 0.05$, ** $P < 0.01$, *n*: total numbers of mosquitoes used in the assays. sgRNA-*AaRel1* lines were used as controls. Created with BioRender.com.

prevalence (percent of infected mosquitoes with at least one virion) also was reduced significantly with 18% and 20% reductions, respectively, in both transactivated lines (**Fig 4E**, $^*$ $P < 0.05$, $^{**}$ $P < 0.01$). The trend of higher median viral loads suppression in the transheterozygotes line A (dCas9-VPR/U6-sgRNA$^{rel1-A}$) compared to line B is statistically insignificant (**Fig 4E**, Mann-Whitney test, $P = 0.886$). The suppression of DENV2 viral infection in terms of both median intensity and prevalence indicates that CRISPRa-mediated transactivation of *AaRel1* in the midgut augmented the activity of the Toll pathway's antiviral action in this tissue, consistent with previously-published work [43] concluding that Toll immune signaling pathway plays a significant role in regulating resistance to dengue virus in the mosquito midguts [43]. RNAi-mediated gene silencing of the *Rel1* transcriptional factor resulted in significant suppression of DENV2 viral loads while silencing of the negative regulator, *Cactus*, resulted in a 3-fold upregulation of the *Rel1* gene and a 3.3-fold decrease of the mean viral titers [43]. Here the mean viral loads combined with both infection intensities and non-infected mosquitoes reached ~5.2 fold reduction. The extent of viral suppression is stronger than that displayed when using RNAi-mediated gene silencing of the Toll pathway negative regulator *Cactus*, most likely due to the more robust immune activation of the Toll immune signaling pathway achieved through CRISPRa-mediated transactivation of the Rel1 transcription factor directly, other than manipulation of the negative regulator through RNAi-mediated gene silencing.

## Discussion

Here we have demonstrated programmable transactivation of endogenous genes in *Ae. aegypti* using a CRISPRa system. As a proof of concept, we targeted two conserved embryonic development genes, *eve* and *hh*. Both qPCR and RNAseq analysis verified significant upregulation of the target genes resulting in increased transcript abundance only within individuals containing all CRISPRa components. In addition, embryonic lethality and transcript mispatterning in the embryo phenotypes were observed and related to overexpression of *eve* and *hh*. It is not surprising that overexpression of these developmental genes results in embryonic lethality as similar phenotypes are observed in *D. melanogaster* using a CRISPRa approach [38].

The binary approach that maintains dCas9-VPR and sgRNAs in separate *Ae. aegypti* transgenic lines allow for further expansion and utility of the CRISPRa system. By separating these key components, a library of dCas9-VPR lines expressed under various promoters as well as sgRNAs targeting different genes can be designed and generated to allow for future flexibility. In this study, we designed a dCas9-VPR line expressed using the polyubiquitin promoter (PUb), which is active in multiple cell types throughout development as well as two lines expressing multiplexed sgRNAs targeting the promoter regions of developmental genes *eve* and *hh*. Additional sgRNA lines targeting other genes can be generated and crossed with the PUb:dCas9-VPR line or additional dCas9-VPR lines generated using different promoters that are spatially- or temporally-specific for more targeted transactivation of genes.

Although we confirmed transactivation of target genes using our system, it is important to note that there is a possibility of off-target effects. dCas9 is capable of binding DNA sequences with as many as nine consecutive mismatches in the PAM- distal region [49]. Therefore, we analyzed each of the predicted off-target binding sites *in silico* and determined that the few off-target genes that resulted in increased transcript accumulation seen in the RNAseq data are important during development and include a homeobox gene. Thus, their altered expression is likely linked to the overexpression of *eve*. It is difficult to determine if the increased expression of these genes is linked to general effects of overexpressing the target genes (*eve* or *hh*), or a result of direct off-targeting by the CRISPRa system. However, we determined from our

analysis that the possibility of true off-target effects are low as were observed in previous studies [45,58,59]. To further understand and confirm transcriptome-wide effects caused from transactivating target genes, a variety of sgRNAs of different sequences could be used. Corroborating RNAseq results between these sgRNAs would strengthen the understanding of which affected non-target genes are expressed differentially due to target gene transactivation versus direct off-targeting. Furthermore, we transactivated two transcription factors known to play pivotal roles within embryonic developmental pathways. Transactivating genes that have no effect on transcription regulation may result in a reduced effect in differential expression within the transcriptome. In future studies, we strongly recommend using one of the existing online sgRNA design tools to minimize off-target binding sites in the genome (reviewed in [60]). Moreover, to minimize potential off-target effects, especially when studying transcription factor genes, additional methods like Chip-seq should be considered.

Overall, our results demonstrate that CRISPRa is viable in *Ae. aegypti* as a means to effectively transactivate specific genes. Current genetic engineering tools for *Ae. aegypti* have focused on knocking down, or out, genes as well as expressing transgenes. Among the available tools, the use of UAS/GAL4 and Q-systems allows expression of transgenic elements in several mosquito tissues [33,34]. However, modifications of GAL4 may be required for stronger expression. The QF2 transactivator of the Q-system has been reliably strong, but some toxicity effects may be present for broad expression patterns [61,62]. Despite the impressive gain in knowledge of molecular genetics of mosquitoes in the last decades [63,64], there has yet to be a tool for the targeted activation of endogenous genes. The ability to transactivate select endogenous genes can lend towards further understanding of *Ae. aegypti* genes through functional studies involving overexpression experiments.

Transactivating endogenous loci is important for evaluating genes that have detrimental effects on development, reproduction, nutritional and metabolic processes, and other aspects of mosquito biology. Furthermore, studying the activation of specific immune pathways in response to virus infection also will help to gain knowledge of mosquito immunity. In addition, the use of the dCas9-VPR system can be employed to further expand the possibilities for vector control tools [38]. Previously, a system in *D. melanogaster* was used to generate synthetic speciation, which could be used to introduce reproductive barriers within wild populations [38]. To this purpose, dCas9-VPR can be directed to target endogenous genes, essential for development and viability, to induce lethality. This lethality can be then rescued in engineered, but not wild-type, individuals through mutations of the target sites as described in the engineering of synthetic SPECIES [38]. Target site mutations will prevent dCas9 binding, and subsequent lethal overexpression, without interfering with target gene function. Thus the system can be used to create reproductive barriers that could be used for ecosystem engineering or pest and vector population control.

The dissection of potent roles of mosquito innate immunity requires powerful genetic tools. Previous studies relied on tissue-specific overexpression of the key regulators or RNAi-mediated silencing of either positive or negative regulators to investigate the role of the immune signaling pathways, Toll, IMD, JAK-STAT, and siRNA. The lack of specificity of RNAi-mediated gene silencing and flexibility and efficiency of tissue-specific transgenic overexpression of these key molecules have hampered the depth of the study of the immune signaling cascades and molecular mechanisms of the antiviral defenses in the mosquitos innate immunity. Transgenically tissue specific overexpressing of these immune signaling pathway regulators has been proven as a powerful approach to study the function of these transcriptional factors. For example, transgenic overexpression of the JAK-STAT positive regulators, DOME and HOP, driven by a vitellogenin gene promoter in the fat body tissue could pinpoint the specific antiviral function of this pathway to different viral species. It's shown the

JAK-STAT immune pathway plays an important role in the suppression of viral infection with DENV, but not with ZIKV or CHIKV [56]. Similarly, the same technology has been applied to study the functions of key factors of the Toll, IMD, and siRNA immune pathways [42,44,55,65]. However, the technical challenges and labor intensive nature of this approach have limited its applications on a broader scope. RNAi-mediated gene silencing turned out to be more flexible and with a fast turnover to give a quick glimpse of the function of the specific genes. The NF-κB-like Relish-like transcription factors, Rel1 for Toll pathway and Rel2 for IMD pathway, play central roles in the mosquito innate immunity, which were both studied through RNAi-mediated gene silencing and transgenically overexpressing in the *Aedes* mosquitoes [41–44,65]. The advantage of using RNAi gene silencing is to allow a transient stimulation of the Toll and Imd pathways in the absence of a microbial elicitor. The activation of these pathways can be achieved through the transcriptional down-regulation of the negative regulator, Cactus or Caspar, for the Toll or Imd pathway, respectively [43]. However, the drawback of this approach is the activation of the immune pathway is an indirect process. The whole genome transcriptome analysis has suggested that the immune activation through indirect down-regulation of the negative regulators through RNAi-mediated gene silencing doesn't necessarily activate the same immune signaling cascades as that obtained from direct overexpression of the positive regulators [44]. The immune transcriptome from *Rel1*-overexpressing transgenic mosquitoes revealed a significant 50% overlap with that obtained from the *Cactus* gene-silenced mosquitoes. In contrast, the *Rel2*-overexpressing transgenic mosquitoes regulated transcriptome showed a relatively smaller overlap with those from *Caspar* (a negative regulator of IMD pathway) gene silenced mosquitoes, suggesting that *Caspar* contributes to the regulation of a small subset of IMD pathway or Rel2-regulated genes.

We show here that CRISPRa-mediated transcription activation of immune factors, such as Toll immune pathway positive regulator Rel1, can be applied to study the role of immune pathway genes in the antiviral defenses. The advantage of this genetic tool is the activation of the key immune transcriptional factors results in more profound upregulation of the transcription factor gene and therefore restricting the viral infection in the mosquitoes more efficiently. Future development of the tissue-specific dCas9-VPR lines in combination with target-specific sgRNA lines will allow key immune factors to be studied in a spatial-temporal manner. This tool can be used to boost mosquito immunity against viral infections by transactivating genes involved in various immune pathways including the Relish-like transcription factors Rel1 and Rel2, two key downstream regulators of the Toll and IMD immune pathways [41,65,66], the insect cytokine-like factor Vago [67,68], DOME and HOP in the JAK-STAT pathway [55,56,69], or Dicer2 and R2d2 in the siRNA immune pathway [55,56,69]. The knowledge gained from these basic studies will further strengthen the development of vector control strategies. Furthermore, this system could also be applied to *Ae. aegypti* to drive select genes into populations as part of a population modification strategy for vector-based disease control.

## Methods

### Ethics statement

All animals were handled in accordance with the Guide for the Care and Use of Laboratory Animals as recommended by the National Institutes of Health and approved by the University of California, San Diego Institutional Animal Care and Use Committee (IACUC, Animal Use Protocol #S17187) and University of California, San Diego Biological Use Authorization (BUA #R2401). The protocol (permit # MO15H144) was approved by the Animal Care and Use Committee of Johns Hopkins University. Commercially obtained anonymous human blood type O+, and untyped human serum (InterState Blood Bank, Inc.) was used for DENV2

infection assays in the mosquitoes. The John Hopkins Bloomberg School of Public Health Committees on Human Research reviewed the use of commercial anonymous human blood for mosquito feeding on April 9 2004 and considered it as not involving human subjects as defined by federal regulations, thus informed consent was not required. The biosafety registration of DENV2 was approved by the Johns Hopkins Institutional Biosafety Committee, HSE (Health, Safety, and Environment).

## Mosquito rearing and maintenance

All *Ae. aegypti* lines used in this study were generated from the Liverpool strain. Colonies were reared at 27.0°C, 20–40% humidity, and a 12-h light/dark cycle. Adults were fed 0.3M aqueous sucrose *ad libitum*. To produce eggs, mature females were blood-fed using anesthetized mice. Oviposition cups were provided ~3 days post blood-meal and eggs were collected and aged for ~4 days before hatching. Matured eggs were submerged under deionized $H_2O$ and placed into a vacuum chamber set to 20 in Hg overnight. Emerged larvae were reared in plastic containers (Sterilite) with ~3 liters of deionized $H_2O$ and fed daily with fish food (Tetramin). *Aedes* mosquitoes rearing at the Johns Hopkins Insectary Core Facility followed established standard procedures and were maintained on 10% sucrose solution under standard insectary conditions at 27±0.5°C and 75–80% humidity with a day:night light cycle of 14:10 hr.

## Construct design and assembly

The Gibson enzymatic assembly method was used to engineer all constructs in this study. To generate the dCas9-VPR expressing construct, OA-986F (Addgene plasmid #183993), Our previously published OA-986A plasmid was used as a backbone [38]. Restriction enzymes, NotI and PmeI, were used to cut the plasmid backbone. DNA fragments containing the PUb promoter (AAEL003877) was amplified from the Addgene plasmid #100581 using primers 874r32 and 986F.C1. In addition, we generated 2 sgRNA plasmids each harboring 2 distinct sgRNAs targeting the promoter regions of *eve* (AAEL007369, OA-1053A, Addgene plasmid #184006) and *hh* (AAEL006708, OA-1053B, Addgene plasmid #184007). To engineer these plasmids, we modified plasmid OA-984 [38] (Addgene plasmid #120363) to contain 2 sgRNA sequences targeting either *eve* or *hh* driven by U6 promoters (sgRNA$^{eve}$ and sgRNA$^{hh}$). Restriction enzymes AvrII and AscI were used to create the plasmid backbone. Two GenPart fragments were synthesized from GenScript for each plasmid (OA-1053A or OA-1053B), one containing sgRNA1 driven by U6b (AAEL017774), while the other containing sgRNA2 driven by U6c (AAEL017763). For virus targeting, we generated one plasmid harboring 4 distinct sgRNAs targeting the promoter region of *AaRel1* (AAEL007696, OA-1127B, Addgene plasmid #190997). Firstly, two intermediate plasmids, OA-1127B.X1 (*rel1*-gRNA1&2) and 1127B.X2 (*rel1*-gRNA3&4), each harboring two gRNAs, were generated by cutting the same previous backbone plasmid OA-984 (Addgene plasmid #120363), with the restriction enzymes AvrII and AscI, and cloning in two GenPart fragments, which were synthesized from GenScript containing two gRNAs driven by U6b (AAEL017774) and U6c (AAEL017763) promoters respectively. Then, plasmid OA-1127B.X1 was linearized with the restriction enzyme FseI, and the insertion of U6b-*rel1*-gRNA3-U6c-*rel1*-gRNA4 was amplified with primers 1167.C5 and 1067. C6 from the plasmid OA-1127B.X2. During each cloning step, single colonies were chosen and cultured in Luria Bertani (LB) Broth with ampicillin. Plasmids were extracted using the Zippy plasmid miniprep kit ((Zymo Research, Cat No./ID: D4036) and sanger sequenced. Final plasmids were maxipreped using the ZymoPURE II Plasmid Maxiprep kit (Zymo Research, Cat No./ID: D4202) and sanger sequenced in preparation for embryonic microinjection. All primers are listed in **S10 Table**. Complete plasmid sequences and plasmid DNA are

available at www.addgene.com with accession numbers (#183993, #100581, #184006, #184007, #120363, #190997).

## Generation of transgenic lines

Transgenic lines were generated by microinjecting 0.5–1 h old pre blastoderm stage embryos with a mixture of the piggyBac plasmid (200 ng/µl) and a transposase helper plasmid (phsp-Pbac, (200 ng/µl) [70]. Embryonic collection and microinjections were performed following previously established procedures. After 4 days post-microinjection, $G_0$ embryos were hatched in deionized $H_2O$ under vacuum (20 in Hg). Emerged larvae were reared to pupal stage using previously established procedures. Surviving $G_0$ pupae were sex-separated into ♀ or ♂ cages. WT ♀ or ♂ of similar age were added to cages of the opposite sex at a 5:1 ratio (WT:$G_0$). Several days post-eclosion (~4–7), a blood-meal was provided, and eggs were collected, aged, then hatched. Hatched larvae were screened and sorted for expression of relevant fluorescent markers using a fluorescent stereo microscope (Leica M165FC). Each individual line was maintained as mixtures of homozygotes and heterozygotes with periodic selective elimination of wild-types.

## Generating and screening for CRISPRa transheterozygotes

PUb-dCas9 and sgRNA females were outcrossed to WT males to create heterozygous progeny that could be used as the parents for the transheterozygote crosses. Resulting progeny positive for their respective fluorescent markers were collected and considered true transgenic heterozygotes. PUb-dCas9 females were crossed with sgRNA males. Reciprocal crosses also were performed. After allowing the mosquitoes to mate for 3 days, females were blood-fed for 2 consecutive days. Three days after blood-feeding, females were individually captured in plastic vials lined with moistened paper. Captured females were kept for 2 days to allow for egg laying and removed afterwards. Collected eggs were either processed for RNA collection, fixed and dechorionated for staining, or hatched to screen surviving progeny.

## Total RNA collection and sequencing

To directly observe and quantify targeted *eve* and *hh* transactivation mediated by PUb:dCas9-VPR, we collected transheterozygous embryos for RNA extraction and subsequent sequencing. Embryos were collected 24 h post-oviposition from F1 transheterozygous lines (PUb:dCas9-VPR/U6-sgRNA$^{eve}$ and PUb:dCas9-VPR/U6-sgRNA$^{hh}$) as well as parental lines (PUb:dCas9-VPR; U6-sgRNA$^{eve}$ and U6-sgRNA$^{hh}$). Three biological replicates per line were collected for a total of 15 samples. Additionally, To quantify the transactivation of *rel1*, one day old adult females were used for RNA extraction and subsequent qPCR analysis. Three biological replicates per line were collected for a total of 15 samples, including the two transheterozygous lines (PUb:dCas9-VPR/U6-sgRNA$^{rel1-A}$, and PUb:dCas9-VPR/U6-sgRNA$^{rel1-B}$) and the three parental controls (PUb:dCas9-VPR, U6-sgRNA$^{rel1-A}$ and U6-sgRNA$^{rel1-B}$). Total RNA was extracted using a Qiagen RNeasy Mini Kit (Qiagen 74104). Following extraction, total RNA was treated with an Invitrogen DNase treatment kit (Invitrogen AM1906). RNA concentration was analyzed using a Nanodrop OneC UV-vis spectrophotometer (Thermo-Fisher NDONEC- W). About 1 µg of RNA was used to synthesize cDNA with a RevertAid H Minus First Strand cDNA Synthesis kit (Thermo Scientific). CDNA was diluted 50 times before use in Real-Time quantitative PCR (RT-qPCR). RT-qPCR was performed with SYBR green (qPCRBIO SyGreen Blue Mix Separate-ROX Cat #: 17-507B, Genesee Scientific). 4 µl of diluted cDNA was used for each 20 µl reaction containing a final primer concentration of 200 nM and 10 µl of SYBR green buffer solution. Three technical replicates for each reaction were

performed to correct for pipetting errors. The following qPCR profile was used on the Light-Cycler instrument (Roche): 3 min of activation phase at 95°C, 40 cycles of 5 s at 95°C, 30 s at 60°C. **S10 Table** lists the primers for *eve*, *hh*, *AaRel1* and *rpl32* (*ribosomal protein L32*). The *rpl32* gene was used as a reference gene [71] to calculate relative expression level of *eve*, *hh* and *AaRel1* with the manufacturer software and the delta-delta Ct method ($2^{-\Delta\Delta Ct}$). Difference in expression of *eve*, *hh* and *AaRel1* between controls and transactivated lines, was statistically tested with one way ANOVA and a Tukey's multiple-comparison test in RStudio statistical software (version 1.2.5033).

Collected RNA for *eve* and *hh* was also used to perform RNA-seq analyses in order to further validate results from qPCR analysis as well as detect other genes affected by *eve* and *hh* upregulation and potential off target genes. RNA integrity was assessed using RNA 6000 Pico Kit for Bioanalyzer (Agilent Technologies 5067–1513) and RNA-seq libraries were constructed using NEBNext Ultra II RNA Library Prep Kit for Illumina (NEB E7770) following manufacturer's instructions. Libraries were sequenced on Illumina HiSeq2500 in single read mode with the read length of 50 nt and sequencing depth of 20 million reads per library. Basecalling was performed with RTA 1.18.64 followed by conversion to FASTQ with bcl2fastq 1.8.4.

### Quantification and differential expression analysis

RNA integrity was assessed using the RNA 6000 Pico Kit for Bioanalyzer (Agilent Technologies #5067–1513), and mRNA was isolated from ~1 μg of total RNA using NEBNext Poly(A) mRNA Magnetic Isolation Module (NEB #E7490). RNA-seq libraries were constructed using the NEBNext Ultra II RNA Library Prep Kit for Illumina (NEB #E7770) following the manufacturer's instructions. Libraries were quantified using a Qubit dsDNA HS Kit (ThermoFisher Scientific #Q32854), and the size distribution was confirmed using a High Sensitivity DNA Kit for Bioanalyzer (Agilent Technologies #5067–4626). Libraries were sequenced on Illumina HiSeq2500 in single read mode with the read length of 50 nt and sequencing depth of 20 million reads per library. Basecalling was performed with RTA 1.18.64 followed by conversion to FASTQ with bcl2fastq 1.8.4. The reads were mapped to *Aedes aegypti* genome AaegL5.0 (GCF_002204515.2) supplemented with PUb-dcas9 transgene sequence using STAR [72]. Gene expression was then quantified using featureCounts against NCBI *Aedes aegypti* Annotation Release 101 (GCF_002204515.2_AaegL5.0_genomic.gtf). TPM values were calculated from counts produced by featureCounts and combined (combined_count_tpm.aaegl5_d-Cas9-vpr.xlsx). Hierarchical clustering and PCA analyses were performed in R and plotted using R package ggplot2. Differential expression analyses between controls (PUb:dCas9-VPR; U6-sgRNA$^{eve}$ and U6-sgRNA$^h$) and transheterozygous lines (PUb:dCas9-VPR/U6-sgRNA$^{eve}$ and PUb:dCas9-VPR/U6-sgRNA$^{hh}$) were performed with DESeq2 (deseq2_sgRNA_transhet_Eve.xlsx, deseq2_dCAS9_VPR_transhet_Eve.xlsx, deseq2_sgRNA_transhet_hh.xlsx, deseq2_dCAS9_VPR_transhet_hh.xlsx). Illumina RNA sequencing data has been deposited to the NCBI-SRA, (accession number PRJNA851480, https://www.ncbi.nlm.nih.gov/bioproject/PRJNA851480.)

### Phenotypic Screening

To collect egg lays from single pair mating events, female and male mosquitoes were allowed to mate for 3 days post eclosion. Females were given a blood meal for 2 consecutive days. The day following the 2nd blood meal, blood-fed females were placed independently into plastic drosophila vials lined with wet paper and plugged with a foam plug. The females were kept in the vials for 2–3 days to allow for egg laying. Following oviposition onto the paper lining the drosophila vial, females were released into a small cage and egg lays were collected, counted,

and allowed to mature to full development (~4 days) in their original vials. Matured eggs were hatched within their original vial under vacuum overnight. Following hatching, egg papers were removed from the vials to allow for more space for the larvae to grow. At the L3 stage, progeny were screened, scored, and counted for expression of opie-2-dsRed and 3xP3-tdTomato using a fluorescent stereoscope (Leica M165FC). The difference in total larval counts compared to total egg counts were considered to be dead during embryonic or early larval stages. Surviving transheterozygous individuals were collected for further observation and analysis.

### *In situ* hybridization and embryo imaging

Embryos 24 h post-oviposition were collected, fixed, and dechorionated using previously described methods [73]. To more effectively remove the endochorion, peeling was performed in a mixture of PBS/PBT instead of methanol/ethanol. To free the embryo from the endochorion, fine tip forceps were used to crack a ring around the middle of the egg, taking care to not puncture the embryo. Cracked embryos were then briefly placed in methanol then back into PBS to improve detachment of the embryo from the endochorion. Each end of the endochorion was then teased off of the embryo. Yolk clarification was then performed according to previously described methods [73]. HCR *in situ* hybridization was performed using previously described methods [74]. HCR probes purchased from Molecular Instruments. Stained embryos were imaged using a Leica SP8 Confocal with Lightning Deconvolution.

### Virus propagation and oral viral infections in *Ae. aegypti*

DENV serotype 2 New Guinea C strain (DENV2) was cultured in *Aedes albopictus* C6/36 cells (ATCC CRL-1660), and viral stocks were prepared as previously described in [55,56,69]. All infection procedures were performed under BSL2 conditions for DENV2. Briefly, C6/36 cells were cultured in MEM medium (Gibco, Thermo Fisher Scientific, USA) supplemented with 10% heat-inactivated fetal bovine serum (FBS), 1% penicillin-streptomycin, and 1% non-essential amino acids and maintained in a tissue culture incubator at 32°C and 5% $CO_2$. Baby hamster kidney strain 21 (BHK-21, ATCC CCL-10) cells were maintained at 37°C and 5% $CO_2$ in the DMEM medium (Gibco, Thermo Fisher Scientific) supplemented with 10% fetal bovine serum (FBS), 1% penicillin-streptomycin, and 5μg/ml Plasmocin (InvivoGen, USA). For the preparation of DENV2 viral stocks, C6/36 cells grown to 80% confluence were infected with DENV2 at a multiplicity of infection (MOI) of 10 and incubated at 32°C and 5% $CO_2$ for 5~6 days. Virus was harvested by three freeze-thaw cycles using dry ice and a water bath (37°C), followed by centrifugation at 2,000 rpm for 10 min at 4°C. The supernatant from this cell lysis was mixed with the original cell culture supernatant to yield the final viral stock. Viral stocks were aliquoted and stored at -80°C for long-term storage. Seven-day-old mosquitoes were orally infected with DENV2 through artificial glass membrane feeders as previously described [55,56,69]. A portion of each blood meal was frozen, and back titrated by plaque assay on the BHK-21 cells at 37°C. Mosquitoes were starved for 24 h prior to being offered the blood meal and were allowed to feed for ~30 min. Fully engorged mosquitoes were sorted into soup cups, with no more than 60 individuals per cup. Each experiment was performed in at least three biological replicates, as indicated.

### Plaque assays for viral titration

DENV2 infected mosquito samples were titrated in the BHK-21 cell culture and plaque assays were used to determine infection prevalence and the viral titers. In brief, mosquito midguts were collected at 7 days post-infectious blood meal (PIBM) in 150 μl of complete DMEM

medium with glass beads. A Bullet Blender (Next Advance, USA) was used to homogenize the tissue samples, and serial dilutions were prepared with DMEM complete medium. The BHK-21 cells were split to give a 1:10 dilution and grown on 24-well plates to 80% confluence 1–2 days before the plaque assays. After serially diluted, the mosquito tissue or viral stock samples (100 μl each) were added to the BHK-21 cells, followed by incubation at room temperature for 15 min on a rocking shaker (VWR International LLC) and subsequent incubation at 37˚C with 5% $CO_2$ in a cell incubator (Thermo Fisher Scientific) for another 45 min. The 24-well plates with infected BHK-21 cells were overlaid with 1 ml of 0.8% methylcellulose in complete DMEM medium with 2% FBS and incubated for 5 days in a cell culture incubator (Thermo Fisher Scientific, 37˚C and 5% $CO_2$). Plaques were fixed and developed with staining reagent (1% crystal violet in 1:1 methanol/acetone solution) at room temperature for 2 h. Plates were rinsed with distilled water and air-dried, and plaques were counted and multiplied by the corresponding dilution factors to calculate the plaque-forming units (PFUs) per sample. Three biological replicates were done with viral infection assays, and three replicates were pooled to generate the final figure. Dot-plot of infection intensities and pie-chart of infection prevalence were prepared with GraphPad Prism 9 software, and the significance of the infection intensities was determined by Mann-Whitney test and infection prevalence by Fisher's exact test. The distributions of the viral titers are non-normal and therefore the median is used to describe the central tendency. A single extreme outlier can significantly skew the mean, but will have little effect on the median. The non-parametric Mann-Whitney test was used to calculate the *p*-values and determine the statistical significance of viral infection intensities.

## Supporting information

**S1 Table. RNA sequencing TPM values.**
(XLSX)

**S2 Table. Summary TPM value for eve and hh.**
(XLSX)

**S3 Table. Differential expression analysis *eve* transheterozygotes vs. U6:sgRNA[eve] control.**
(XLSX)

**S4 Table. Differential expression analysis *hh* transheterozygotes vs. U6:sgRNA[hh] control.**
(XLSX)

**S5 Table. Differential expression analysis *eve* transheterozygotes vs. PUb:dCas9-VPR control.**
(XLSX)

**S6 Table. Differential expression analysis *hh* transheterozygotes vs. PUb:dCas9-VPR control.**
(XLSX)

**S7 Table. Summary of gene differentially expressed in the different treatments.**
(XLSX)

**S8 Table. Off target analysis *eve*.**
(XLSX)

**S9 Table. Off target analysis *hh*.**
(XLSX)

**S10 Table. Primers and sgRNA sequences used in this study.**
(XLSX)

**S1 Fig. Integrative Genomics Viewer (IGV) Snapshot of the RNAseq data for *eve* overexpression.**
(PNG)

**S2 Fig. Integrative Genomics Viewer (IGV) Snapshot of the RNAseq data for *hh* overexpression.**
(PNG)

**S3 Fig. Larval qPCR.**
(TIFF)

## Acknowledgments

We thank Judy Ishikawa for mosquito husbandry assistance. We thank the Johns Hopkins Malaria Research Institute Insectary for providing the mosquito-rearing facility and the Parasitology Core facilities for providing the naïve human blood.

The views, opinions, and/or findings expressed are those of the authors and should not be interpreted as representing the official views or policies of the U.S. government.

## Author Contributions

**Conceptualization:** Anthony A. James, Michael W. Perry, George Dimopoulos, Omar S. Akbari.

**Data curation:** Michelle Bui, Elena Dalla Benetta, Yuemei Dong, Yunchong Zhao, Ting Yang, Ming Li, Anna Buchman, Vanessa Bottino-Rojas.

**Formal analysis:** Michelle Bui, Elena Dalla Benetta, Yuemei Dong, Yunchong Zhao, Ting Yang, Ming Li, Anna Buchman, Vanessa Bottino-Rojas.

**Funding acquisition:** Anthony A. James, George Dimopoulos, Omar S. Akbari.

**Investigation:** Michelle Bui, Elena Dalla Benetta, Yuemei Dong, Yunchong Zhao, Ting Yang, Ming Li, Igor A. Antoshechkin, Anna Buchman, Vanessa Bottino-Rojas.

**Methodology:** Michelle Bui, Elena Dalla Benetta, Yuemei Dong, Yunchong Zhao, Ting Yang, Ming Li, Igor A. Antoshechkin, Anna Buchman, Vanessa Bottino-Rojas.

**Project administration:** Anthony A. James, Michael W. Perry, George Dimopoulos, Omar S. Akbari.

**Supervision:** Anthony A. James, George Dimopoulos, Omar S. Akbari.

**Validation:** Michelle Bui, Elena Dalla Benetta, Yuemei Dong, Yunchong Zhao, Ting Yang, Ming Li, Igor A. Antoshechkin, Anna Buchman, Vanessa Bottino-Rojas.

**Visualization:** Michelle Bui, Elena Dalla Benetta, Yuemei Dong, Yunchong Zhao, Ting Yang, Ming Li, Igor A. Antoshechkin, Anna Buchman, Vanessa Bottino-Rojas.

**Writing – original draft:** Anthony A. James, George Dimopoulos, Omar S. Akbari.

**Writing – review & editing:** Michelle Bui, Elena Dalla Benetta, Yuemei Dong, Yunchong Zhao, Ting Yang, Ming Li, Igor A. Antoshechkin, Anna Buchman, Vanessa Bottino-Rojas, Anthony A. James, Michael W. Perry, George Dimopoulos, Omar S. Akbari.

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
