## [Decision Letter · Decision Letter 0]

28 Nov 2022

Dear Dr. Akbari,

Thank you very much for submitting your manuscript "CRISPR Mediated Transactivation in the Human Disease Vector Aedes aegypti" for consideration at PLOS Pathogens. As with all papers reviewed by the journal, your manuscript was reviewed by members of the editorial board and by several independent reviewers. The reviewers appreciated the attention to an important topic. Based on the reviews, we are likely to accept this manuscript for publication, providing that you modify the manuscript according to the review recommendations.

As you will see, two of the reviewers had minor comments, one had major comments. While there is no additional experimental work required, there is substantial rewriting needed, I think. Two of the reviewers questioned the novelty of the work and so you may wish to address that in the rewrite as well. Please attempt to address all of the reviewer comments.

Sincerely,

Elizabeth A. McGraw, PhD

Academic Editor

PLOS Pathogens

Kami Kim

Section Editor

PLOS Pathogens

Kasturi Haldar

Editor-in-Chief

PLOS Pathogens

orcid.org/0000-0001-5065-158X

Michael Malim

Editor-in-Chief

PLOS Pathogens

orcid.org/0000-0002-7699-2064

As you will see, two of the reviewers had minor comments, one had major comments. While there is no additional experimental work required, there is substantial rewriting needed, I think. Two of the reviewers questioned the novelty of the work and so you may wish to address that in the rewrite as well. Please attempt to address all of the reviewer comments.

Reviewer Comments (if any, and for reference):

Reviewer's Responses to Questions

**Part I - Summary**

Reviewer #1: see attached file

Reviewer #2: This reports describes the use of the CRISPRa system ( a 'dead' but guidable version of CRISPR-Cas9 fused to transcriptional activator) as a way to activate the expression of endogenous genes in the mosquito Aedes aegypti.

CRISPRa has been well used in many other systems, including the model insect Drosophila, but this is the first report of its use in a disease vector. Experiments are performed technically well, and I'm happy to accept that there is specific upregulation of genes targeted by CRISPRa.

The case for how this is enabling for future vector control applications however, could be better made. As could the case for how it is a step change over existing transactivation of transgenes through Gal4 or tTA etc, which are not really mentioned. Yes this is upregulating an endogenous gene, but tell us why this matters.

In terms of the effects of CRISPRa activation, there are many global transcriptome changes, as would be expected from the misexpression of important developmental transcription factors but I'm not sure how much insight is added by these analyses. By upregulating an immunity transcription there is a real, but modest, effect on Dengue titre and prevalence. Again, the case for where this will lead in any application could be made more strongly.

Reviewer #3: In this manuscript, Bui et al describe their work on the development in Aedes aegypti, an important vector of arboviral diseases, of the CRISPa transactivation system for the overexpression/misexpression of endogenous genes. This system has previously been built in Drosophila and has been used both for basic research and for applications of vector control, as mentioned in this manuscript. The Akbari lab is a leader in the engineering of this mosquito, and other insects and the works build nicely on the previous work of this team. The authors clearly demonstrate that the system works as expected, based on the detection of target gene misexpression and, depending on the target gene, downstream misexpression of downstream targets affecting development and immunity.

In terms of novelty, this system was not previously available in this important insect, and now it is - whether that is sufficient novelty is questionable to this reviewer - but then novelty is partially subjective and so is usefulness. It is clear that the authors intended this mainly as a tool development paper demonstrating proof-of-concept, which they have succeeded in doing.

Having said that, the design of the study isn't especially detailed or exploratory of the parameters of the system. Therefore, questions of design of future research will have to be addressed independently of this work - i.e. is this work a future reference for how to best use the CRISPa system in aegypti - probably not. Just that it can work. For example, the work describes two construct designs for expression of sgRNAs, one set has two sgRNA expression cassettes (used for eve and hh) and the second uses two sets of two sgRNAs. The authors state that they do this to increase efficiency - but if this does indeed increase efficiency can’t really be known from their work. While this is in principle understandable - not every design parameter can or should be measurable and/or controlled, this manuscript is packaged as a tool development paper - but how this tool works and can be put together in future work is barely explored.

Another good example of this is the choice in targets: in the first part of the paper they target hh and eve, endogenous transcription factors that regulate the expression of many important developmental genes. As expected, overexpression of either gene is lethal to transgenic individuals, which can be leveraged in applications of engineered genetic incompatibility between two strains (i.e. synthetic species). The authors and other groups have previously developed CRISPa based genetic incompatibility by missexpression of developmental genes in Drosophila, indeed this forms the first half of the incompatibility - the second consists of mutant alleles that retain gene function but are not targetable by dCas9-VPR. Therefore the first part of this manuscript - the encoding of eve/hh based lethality thorough transactivation - does represent a direct step in the process of engineering such genetic control systems. Unfortunately, the choice of these target genes makes characterizing the CRISPa system in general, difficult since transactivation is expected to lead to dominant lethality. It is also difficult to evaluate off-targeting of a dsCas9-VPR transcription factor when the target genes are themselves transcription factors that in turn activate expression of a number of genes themselves. I think with some improvements to the presentation of the work and sharpening of the narrative this work should be published, hopefully to open further exploration of this tool.

**Part II – Major Issues: Key Experiments Required for Acceptance**

Reviewer #1: No major issues

Reviewer #2: In Fig.4C is the difference in expression between line A and B here significant? Both lines have similar (mild) efficacy in reducing viral titre or prevalence of infections. Can we assume then that it is pretty insensitive to expression levels of Rel1? Warrants some discussion.

Four gRNAs were used in the CRISPRa Rel1 set up. Is the effect expected to be cumulative - i.e. more gRNAs more activation?

In Fig2B were the were crosses 2 and 3 compared against cross 1 to see if significant i.e. is II (VPR only) causing significant overexpression without gRNA? If so, are there overlap between these genes up-regulated and those upregulated in the presence of gRNA?

The last two concluding statements in the Discussion are so vague I doubt that many will be able to understand how these findings strengthen or enable these aspects

"The extent of viral

suppression is stronger than that displayed when using RNAi-mediated gene silencing of the Toll

pathway negative regulator Cactus, most likely due to the more robust immune activation of the

Toll immune signaling pathway achieved through CRISPRa-mediated transactivation of the Rel1

transcription factor." - is there RNAi data for the targeting of Rel1 transcription factor directly (rather than Cactus)?

I was interested in the finding that the parental source of the dCas9 or the gRNA made a difference - I suspect this is due to the larger dose of guideRNA that gets into the embryo when inherited maternally rather than any paternal effect associated with delivery of dCAS9. However, none of this was elaborated in the discussion and what it might mean operationally. Also, in later crosses with the Rel1 i assume you do not have the corresponding reciprocal crosses, as above, to see if these difference in parental provision of alleles consistently makes a difference?

"We then searched to see if any nearby gene was

upregulated in our RNAseq data. Among 85 potential off-target sites for the eve or hh sgRNAs,

only 8 sites correlated to eve sgRNAs, localized near genes that were differentially expressed in

our analysis with LogFC > 2 and FDR < 0.05" - syntax and use of (or not) comma are important here. You mean to say 8 out of 85 potential off target sites showed upregulation of nearby genes that would be consistent with off-target binding of the transactivator? Do you have controls here for comparison? if 10% of genes near potential off target sites are differentially regulated, what is the overall percentage of genes differentially regulated? Are all 85 off targets sites near enough to genes to be expected to show differential regulation if they were bound by CRISPRa?

Reviewer #3: 1)

In results 2: Overexpression of eve and hh generated additional transcriptomic changes.

First of all, a comment on the style of narrative. It is written in a difficult to read way, with too many numbers mentioned, and obvious issues glanced over. This reviewer had to read this section multiple times to really understand it, and then discovered what seem to be an important issue - that lead authors likely missed because of how this section is written: 2nd sentence: When reporting on differential expression between transheterozygous and parental strains the authors note that 6488 and 7816 genes are differentially expressed, depending on which of the parental strains are compared - that's around 10% difference - a huge amount. What the (43%) are noted in this sentence is not clear to this reviewer. This difference (and dry reading) continues when another filter is added (logFC > -2|2), although the differences between the two controls becomes smaller. How is it possible that there is such a difference between the controls. This becomes especially relevant perhaps in the ability to detect off-targeting, or sgRNA-independent dCas9-VPR binding and expression. Volcano plots are wholly impractical and insufficient here to describe what is happening, and above half a page of dry information on the number of genes could be summarized in a table or some better plots that properly explore expression including in controls and their appropriate application. The screen-shoting of IGV alignments of the data is amateur, as is the description of the data - for each of the developmental genes targeted, eve and hh, the final sentence of each paragraph describing the RNAseq “analysis” ends with a list of the “most significantly upregulated genes” ordered by ascending significance score. This is based on basic misunderstanding of differential expression - the genes with the lowest significant scores are not the most interesting/significant genes - they are the genes that the algorithm feels most “confident” are differentially distributed in the two datasets. Indeed by looking at the volcano plots, eve and hh are not even themselves the genes with the lowest adjusted p-value.

2)

In the same results chapter: the off-targeting analysis is unclear, including because the methods section misses this part of the work. 62 potential off-targets were identified (although how or what is considered an off-target isn't explained) in the genome and the authors then “searched to see if any nearby gene was upregulated”... what is nearby? how are differences quantified? This section needs to be revised and explained appropriately.

3)

In the next section describing lethality/toxicity there is no follow-up of the transhets that survive, besides looking at them under the microscope to look for morphological differences. This is not enough. What is the basis for their survival - is this heritable?

**Part III – Minor Issues: Editorial and Data Presentation Modifications**

Reviewer #1: see attached file

Reviewer #2: see attached annotated manuscript for these - it is a lot easier than listing them here.

Figures are generally well made, informative and easy to read - a couple of formatting suggestions there

Reviewer #3: I focus mainly on the results section, given time constraints despite the extent of improvements that are needed

Section 1: Lines

Which polyubiquitin gene did you use the promoter of?

Since you cant actually prove that having multiple sgRNAs increases efficacy, provide provide references for where this was shown, or reduce emphasis.

Rephrase “two different regions within the promoter region” to “two different sgRNA sites within 250 bp from the TSS”. I initially read this as distinct regions that were selected based on some criteria.

Fig 1 is very pretty, but in Fig1c its very hard to see the eye fluorescent phenotypes.

Section 2: Overexpression of target genes

Explain why 24hr embryos were selected

The final paragraph describing the transcriptomic data is difficult to read and could be much better described in a figure with many of the details going to the methods. The IGV screenshot should be removed. Fold increases of expression are stated but it is not clear in reference to what? I would suggest that a separate bar plot is made depicting TPMs for eve and hh specifically, in addition to dCAs9 and some control for each of the F1 genotypes, similarly to the Fig 1B, showing the qRT-PCR data.

It looks to me from looking at the TPM values that there are also big differences in hh expression between parental strains. Given that these genes are developmental, I wonder how the design of the study compromised the clarity of the results and this is hard to understand from the results as they are presented.

Section 3: additional transcriptomic changes

See my major comments above

What is 43%? And 41% in eve and hh respectively

Fig2B should be Fig2C

Volcano plots show only one control but the results report important differences

Off-target analysis is unclear (see above)

Reporting on the top genes based on significance is not appropriate. Instead the authors should at least attempt to categorize these genes based on whether they are known or not as being downstream targets of eve or hh

Off-targets that are within 250 bp of TSS should be relevant only in theory, but i dont see how the position of the off-target in relation to the candidate gene is linked.

The sentence “ Furthermore, the probability of off-target transactivation with dCas9-VPR is likely low” should be referenced, or deleted.

Section 4: Lethality

Fig3A does not show what is mentioned in the second sentence.

I have issues with Fig3A in general - pictures of larvae to say that nothing was observed is not ok.

Rephrase: “targeting both genes…”. This can mislead readers to think that a cross between the two sgRNAs, to make individuals containing both were made.

Again difficult to read what the results are, because they are buried in details in whether the cas9-vpr was paternal or maternal in origin. Given this, results are mentioned but their significance cannot be extracted easily.

The in situs that were done to explain the lethality are pretty unimpressive and lacking controls. Sure I will take your word for it. Why were in situs done at this late stage of embryonic development, when the pair rule gene pattern is no longer clearly visible.

Section 5: Rel1 overexpression:

Why did authors choose to increase yet again the number of sgRNAs?

Why is the difference between the transhets and the controls, in terms of median viral titers described twice, once as a 3-fold reduction and once as a 18% reduction. I am confused on the difference between these sentences.

Please rephrase the conclusion of your result here from : consistent with previously published conclusion” to what that conclusion actually was.

It is really a shame that no followup of this was mentioned - feels like the minimum published result of regulating gene expression for immunity engineering.

Other instances:

Reference 71 is repeated several times, and I believe some additional important references need to be added here from other labs working on the same system, not just the outputs of this group.

PLOS authors have the option to publish the peer review history of their article (what does this mean?). If published, this will include your full peer review and any attached files.

Reviewer #1: **Yes: **Michael Smanski

Reviewer #2: No

Reviewer #3: No

Figure Files:

Data Requirements:

Reproducibility:

References:

---

## [Editor Report · Decision Letter 1]

28 Dec 2022

Dear Dr. Akbari,

We are pleased to inform you that your manuscript 'CRISPR Mediated Transactivation in the Human Disease Vector Aedes aegypti' has been provisionally accepted for publication in PLOS Pathogens.

Best regards,

Elizabeth A. McGraw, PhD

Academic Editor

PLOS Pathogens

Kami Kim

Section Editor

PLOS Pathogens

Kasturi Haldar

Editor-in-Chief

PLOS Pathogens

orcid.org/0000-0001-5065-158X

Michael Malim

Editor-in-Chief

PLOS Pathogens

orcid.org/0000-0002-7699-2064
---

## [Editor Report · Acceptance letter]

11 Jan 2023

Dear Dr. Akbari,

We are delighted to inform you that your manuscript, "CRISPR Mediated Transactivation in the Human Disease Vector *Aedes aegypti*," has been formally accepted for publication in PLOS Pathogens.

Best regards,

Kasturi Haldar

Editor-in-Chief

PLOS Pathogens

orcid.org/0000-0001-5065-158X

Michael Malim

Editor-in-Chief

PLOS Pathogens

orcid.org/0000-0002-7699-2064